# WHITE MATTER MATTERS: New Approach to the Brain’s Hidden Half Using Circulating Oligodendrocyte-Derived Extracellular Vesicles

**DOI:** 10.3390/cells14221771

**Published:** 2025-11-12

**Authors:** Masato Mitsuhashi, Dennis Van Epps, Haiping Sun, Li Xing, Keisuke Kawata, Viviana Jimenez, Vernon B. Williams, Cina Sasannejad, Michael L. James, Matthew A. Edwardson, Takuya Murata

**Affiliations:** 1NanoSomiX, Inc., Irvine, CA 92618, USA; dvanepps@nanosomix.com; 2Irvine Materials Research Institute, University of California—Irvine, Irvine, CA 92697, USA; haipins@uci.edu (H.S.);; 3Kinesiology, School of Public Health, Indiana University, Bloomington, IN 47405, USA; 4Center for Sports Neurology & Pain Medicine, Cedars-Sinai Kerlan-Jobe Institute, Los Angeles, CA 90045, USA; 5Department of Neurology, Duke University, Durham, NC 27710, USA; 6Duke Clinical Research Institute, Duke University, Durham, NC 27710, USA; 7Neurology & Rehabilitation Medicine, Georgetown University, Washington, DC 20007, USA; 8Department of Obstetrics and Gynecology, Kawasaki Medical School General Medical Center, Okayama 700-8505, Japan

**Keywords:** white matter, oligodendrocytes, extracellular vesicles, blood test

## Abstract

**Highlights:**

**What is the main finding?**

**What is the implication of the main finding?**

**Abstract:**

White matter, comprising 60% of the human brain, is formed by axonal fibers supported by oligodendrocytes. It is essential for brain communication, yet damage can accumulate silently leading to severe neurological problems. Current diagnostics detect changes only after symptoms appear. To enable earlier detection damage, we developed a blood test monitoring changes in oligodendrocyte-derived extracellular vesicles (ODEs) released from the brain into circulation. After validating the assay, we have shown that ODE levels vary from different individuals. However, ODE levels remain stable under mild head impacts in soccer heading practice (*n* = 15) and boxing/mixed martial arts (*n* = 10), whereas change markedly following neurological insults such as hemorrhagic (*n* = 7) and ischemic stroke (*n* = 14), or gynecological cancer after chemotherapy (*n* = 11). ODE measurement can potentially provide a minimally invasive window into white matter health and support early diagnosis, personalized assessment, and new insights into human brain biology.

## 1. Introduction

The human brain is composed of gray matter and white matter. Gray matter represents about 40% of brain volume and consists of neuronal cell bodies where signals are generated. White matter accounts for the remaining 60% and is formed by axonal fibers supported by oligodendrocytes (OLs). Gray matter functions as the brain’s central processing unit (CPU) while white matter serves as the connecting wires and cables. As a result, white matter has historically received less attention and been studied less extensively than gray matter. Gray matter dysfunction often produces observable symptoms, but white matter injury can remain hidden and neurons may reroute signals through alternative pathways, masking early deficits. Such silent damage accumulates over time and ultimately contributes to severe neurological decline. Detecting these changes before symptoms appear could allow emerging pharmacological [1,2,3,4] and experimental [5,6,7] therapies to slow or even reverse progression. Definitive diagnosis of white matter damage remains possible only at autopsy. Conventional magnetic resonance imaging (MRI) identifies gross structural abnormalities but cannot resolve microstructural integrity. More advanced modalities, such as functional MRI (fMRI) [8] and diffusion MRI (dMRI) [9], offer insights for research but are costly and not widely available in routine practice. Blood biomarkers of OL-specific myelin basic protein (MBP), neuron-specific neurofilament light chain (NFL) and ubiquitin carboxyl-terminal hydrolase L1 (UCHL-1), as well as astrocyte-specific glial fibrillary acidic protein (GFAP) have been extensively studied [10,11]. However, these are intracellular proteins and appear in blood only when cells are severely damaged. Because OLs lie deep in the brain, accessing them requires an invasive biopsy involving drilling through the skull.

Here, we introduce a novel approach for assessing white matter. OLs naturally release extracellular vesicles (EVs)—nanometer-scale particles that carry molecular signatures of their parent cells and enter the bloodstream. EVs include exosomes (30–150 nm), ectosomes (100–1000 nm), and apoptotic bodies (500–2000 nm). Exosomes and ectosomes sharing common biomarkers and comprising the vesicles were analyzed here. By isolating and characterizing OL-derived EVs (ODEs), we have established a minimally invasive blood-based approach for monitoring white matter health and injury. This in turn can enable earlier detection, personalized assessment, and a new avenue for advancing human brain research.

## 2. Materials and Methods

### 2.1. Reagents

All reagents used in this study were consistent with those detailed in our previous publications [12,13,14,15], except for monoclonal antibody against human myelin oligodendrocyte glycoprotein (MOG) (Thermo Fisher Scientific, Waltham, MA, USA). MOG is a type I transmembrane glycoprotein expressed on the surface of OLs and myelin sheath. The antibody recognizes the extracellular domain of the MOG molecule to capture native ODEs without lysis or permeabilization. All antibodies were commercially sourced and underwent appropriate validation through Western blot, immunohistochemistry, flow cytometry, and/or enzyme-linked immunosorbent assay (ELISA) per manufacture. Streptavidin-horseradish peroxidase and chemiluminescent substrate (SuperSignal) were purchased from Thermo Fisher. Biotinylation of antibodies was performed using EZ-Link Sulfo-NHS-LC-Biotin (Thermo Fisher Scientific), followed by a spin column procedure to remove unbound biotin. Recombinant MOG (rMOG) (MedChemExpress, Monmouth Junction, NJ, USA), streptavidin-conjugated 20 nm gold nanoparticles (nanoComposix, San Diego, CA, USA) were purchased from designated suppliers.

### 2.2. Assay Principle

The assay was based on a sandwich chemiluminescent ELISA using a combination of capture and biotinylated detection antibodies, followed by streptavidin-horseradish peroxidase and chemiluminescent substrate reaction. Detailed protocols have been described previously [12,13,14,15]. To indicate the probe-target pairing, we have used the notation “[probe on capture antibody].” For example, when a CD9 probe was applied to EVs captured on anti-MOG–coated wells, we designated it as [CD9 on MOG]. The 96-well ELISA tests generate 96 data points automatically, and each raw value was recorded, minimizing extraneous sample testing variables.

### 2.3. Assessment of EVs

Nanoparticle tracking analysis (NTA), Western blot Scanning, and scanning electron microscopy (SEM) were performed by Particle Technology Labs (Downers Grove, IL, USA), RayBiotech (Peachtree Corners, GA, USA), and University of California Irvine Materials Research Institute (Irvine, CA, USA).

### 2.4. Plasma Samples

Healthy adult plasma samples were purchased from 3 different commercial sources (Innovative Research, Novi, MI, BioIVT, Westbury, NY, USA and Equitech Enterprise, Kerrville, TX, USA). Blood samples from soccer heading practice, provided by Indiana University, were collected before and at 2, 24, and 72 h after extensive heading practice (*n* = 15, all male, no one showed concussion). Blood samples from professional boxers and mixed martial arts (MMA) fighters were collected by NanoSomiX before and at 1, 7, 14, and 35 days after the bout, with some dropouts (*n* = 10, all male, including two fighters who were knocked out by head punches and showed concussion symptoms only on the following day; all others remained asymptomatic). Blood samples from white matter hemorrhagic stroke, provided by Duke University, were collected at 1, 3, 5, and 7 days post-stroke, with some dropouts (*n* = 7; 2 male, 5 female; mean age 57 ± 7.7 years). Blood samples from white matter ischemic stroke, provided by Georgetown University, were collected at 5, 15, and 30 days post-stroke. Blood samples from cancer patients, provided by Kawasaki Medical School, were collected before and after chemotherapy (*n* = 11, all female; 6 ovarian, 4 breast, 1 vaginal; age 39–83 years, mean 64 ± 14). Since this was a feasibility study, post-chemotherapy blood was collected whenever possible. Since we confirmed that plasma samples from venous and capillary blood were equivalent, capillary blood was used for boxing and MMA studies, while venous blood was used for all others(see Table 1). All studies involving human blood samples had institutional review board (IRB) approval and subject consent for research use.

## 3. Results

### 3.1. Validation of ODEs

#### 3.1.1. Validation of EVs

Human control plasma was applied to ELISA wells pre-coated with antibodies against human MOG. To elute captured EVs, pH 2.5 is needed followed by immediate neutralization (Figure 1A, inset). NTA demonstrated that particles had an average size of 112.4 ± 50.2 nm and a density of 1.62 × 10^6^ particles per well, consistent with the size range of exosomes and ectosomes. Eluted and neutralized EVs did not re-bind to anti-MOG wells, suggesting structural damage by low pH exposure. According to the Minimal Information for Studies of Extracellular Vesicles (MISEV) guidelines [17], EV analysis typically involves elution to confirm their presence and quantity of nanoparticles. However, elution may compromise the integrity of the EVs.

#### 3.1.2. Validation of MOG

To avoid elution, captured EVs were directly lysed in ELISA wells and applied to Western blot analysis. As shown in Figure 1B, MOG was identified with a band pattern comparable to mouse brain extracts. Notably, peaks corresponding to dimer-, trimer-, and tetramer-sized species were also observed, consistent with the known tendency of MOG to form such polymers [18].

#### 3.1.3. Visualization of ODEs

The bottoms of ELISA wells were excised, then applied to SEM analysis. As shown in Figure 1C left-middle panels, SEM revealed 100–200 nm spherical structures, confirming EV capture. Furthermore, 20 nm gold nanoparticles targeting CD9 and MOG were observed on the EV surface as bright white spots (Figure 1C right top (anti-CD9) and bottom (anti-MOG)). CD9 is a widely recognized EV marker for both exosomes and ectosomes. These visual data are confirmation of MOG^+^ EVs on the ELISA wells.

### 3.2. Validation of Sandwich Immunoassay

#### 3.2.1. Standard and Specificity

We identified the high-titer plasma sample and assigned it a value of 100 units/mL (U/mL). Figure 2A shows a representative dilution curve, with U/mL on the X-axis and ELISA readings of relative light units (RLU) of [CD9 on MOG] on the Y-axis, where CD9^+^MOG^+^ double-positive signals represent ODEs. Soluble MOG present in plasma does not show signals due to the lack of CD9. Seven plasma samples produced higher signals than the phosphate-buffered saline (PBS) control in anti-MOG wells, whereas all signals were very low in control IgG-wells (Figure 2B).

#### 3.2.2. MOG-Specificity

MOG-specificity of anti-MOG was validated by commercial source using both immunohistochemical staining and Western blot. Because plasma contains various enzymes, we did not add rMOG directly to plasma samples to block MOG^+^ EV binding to anti-MOG wells. Instead, after capturing MOG^+^ EVs on the wells, we applied anti-MOG probes with or without rMOG, since the probe solution lacked proteases. The reaction was dose-dependent, and the presence of rMOG inhibited binding, confirming MOG-specific detection in our assay (Figure 2C).

#### 3.2.3. Enrichment of MOG^+^ EVs

To assess the enrichment of MOG^+^ EVs, plasma samples were applied to wells coated with either anti-CD81 (a common EV marker) or anti-MOG to capture total EVs and MOG^+^ EVs, respectively. Detection was performed using anti-CD81 for total EVs and anti-MOG for MOG^+^ EVs. The ratio of MOG^+^ EVs to total EVs was then calculated. As shown in Figure 2D, MOG^+^ EV content was over sevenfold higher in anti-MOG wells than in anti-CD81 wells, demonstrating selective enrichment.

#### 3.2.4. EV Removal

Two plasma samples were first applied to anti-CD81 wells to capture total EVs, and the supernatants were subsequently transferred to a second set of anti-CD81 wells. As shown in Figure 2E, this procedure reduced the amount of CD81^+^ EVs ([CD9 on CD81]). The same supernatants were then applied to anti-MOG wells and probed with CD9 to quantify ODEs. As shown in Figure 2F, ODE quantities decreased following CD81 depletion, indicating that MOG signals are associated with CD81^+^ EVs.

#### 3.2.5. MOG Removal

Five plasma samples were first applied to anti-MOG wells to capture MOG^+^ EVs, and the supernatants were then transferred to a second set of anti-MOG wells. As shown in Figure 2H, this procedure reduced the amounts of ODEs ([MOG on MOG]). The same samples were probed with anti-CD9 to quantify MOG^+^ EVs. As shown in Figure 2G, MOG^+^ EV levels ([CD9 on MOG]) were also reduced, confirming that MOG is associated with CD9^+^ EVs.

#### 3.2.6. Gold Standard of the Assay

Purified MOG^+^ EVs with known quantities could, in principle, serve as a gold standard for the assay. However, as described in Figure 1A, EVs eluted from anti-MOG wells lost the ability to re-bind. It is not feasible to synthesize recombinant proteins carrying both MOG and CD9 moieties with binding characteristics identical to native MOG^+^ EVs. To address this, various volumes of a standard plasma were spiked into two different plasma samples (Figure 2I). Although these samples had different baseline levels of [CD9 on MOG], the measured values increased proportionally with the standard plasma dilution curve, indicating that the standard plasma can serve as a reliable gold standard for the assay.

### 3.3. Results of Pilot Clinical Feasibility Studies

As shown in Figure 2B, plasma levels of ODEs varied widely across individuals, making this test less suitable for cross-sectional comparisons between disease and control groups at this time. Consistent with our previous observations for neuron-derived EVs (NDEs) [15], we found that, despite high inter-individual variability, intra-individual levels remain stable over time and change markedly in response to specific events. Accordingly, we applied this test to longitudinal plasma samples from subjects of potential white matter damage, including individuals with head trauma (Figure 3A,B), hemorrhagic (Figure 3C) or ischemic stroke (Figure 3D,E) affecting white matter, and post-chemotherapy cancer patients (Figure 3F,G). Chemotherapy was included because it is known to cause greater injury to white matter than gray matter [19]. Since baseline values were largely differed among subjects, % change from the 1st blood samples was evaluated and are shown on the Y-axis in Figure 3. ODE analysis was performed after all samples had been collected and no subsequent interventions were introduced to subjects based on this testing. Notably, plasma volumes as small as 5 µL per well were sufficient for these studies. Since the purpose of this feasibility study was to show the pattern of ODE changes in each subject and that change could be detected, no statistical analysis was performed.

#### 3.3.1. Soccer Heading Practice

Extensive heading practice may induce neurological consequences by stretching, compressing, and twisting axons. As shown previously [16], we aimed to assess acute-phase responses, collecting plasma samples before and up to 72 h after practice. A key strength of this study was the availability of baseline samples from each athlete prior to heading. None of the athletes experienced a concussion. As shown in Figure 3A, plasma ODE levels ([CD9 on MOG]) remained highly stable, suggesting that the heading practice did not cause detectable white matter or OL injury over this 3-day period.

#### 3.3.2. Boxing and MMA

Professional boxing and MMA carry a high risk of concussion. Additionally, fighters are required to arrive at the venue one day before the bout for weigh-ins, allowing blood collection for baseline measurements. Plasma samples were collected before the bout and at 1, 7, 14, and 35 days afterward, extending our previous soccer heading study to a longer 5-week period. Among 10 fighters (all male), two experienced knockouts from clean head punches and showed concussion symptoms only on the following day; all others remained asymptomatic. As shown in Figure 3B, plasma ODE levels ([CD9 on MOG]) remained largely stable throughout the observation period.

#### 3.3.3. Hemorrhagic Stroke

The aim of this study was to evaluate acute-phase biomarkers after hemorrhagic stroke, with blood samples collected during the first week post-event. Pre-stroke samples were unavailable, as patients were only seen after the stroke. We focused exclusively on patients with white matter lesions. Unlike the stable ODE levels observed in Figure 3A,B, some patients exhibited a gradual increase in ODE levels, suggesting that OLs may be activated by the injury. Such an increase may suggest the initiation of a repair response. However, blood samples beyond Day 7 were not available for longer term follow-up. Although conventional MRI was performed at admission, no follow-up MRI was available. Due to the clinical heterogeneity, we present only [CD9 on MOG] results here. Detailed clinical correlations will be reported elsewhere.

#### 3.3.4. Ischemic Stroke

The aim of this study was to evaluate injury and repair-related biomarkers after ischemic stroke, with blood samples collected during the subacute phase at 5, 15, and 30 days post-event. These samples were previously used and published [13]. As with hemorrhagic stroke, pre-stroke samples were unavailable. We focused on patients with subcortical ischemic white matter lesions. As shown in Figure 3D, plasma ODE levels ([CD9 on MOG]) remained stable from Day 5 to Day 30 for most patients. However, Figure 3E highlights three patients with substantial decreases in ODEs. Due to the absence of pre-stroke values, we do not know whether Day 5 values are elevated or not, but substantial decrease from Day 5 to Day 30 may indicate greater white matter injury. Although conventional MRI was performed at admission, no follow-up MRI was obtained. Due to clinical heterogeneity, only [CD9 on MOG] results are presented here. Detailed clinical correlations will be reported elsewhere.

#### 3.3.5. Chemo Brain

This feasibility study aimed to identify blood biomarkers for assessing chemotherapy-induced brain injury, often referred to as “chemo brain”. Plasma samples were collected from cancer patients (*n* = 11, all female; 6 ovarian, 4 breast, 1 vaginal; age 39–83 years, mean 64 ± 14) before chemotherapy and at variable time points afterward (Day 14 to Day 196), whenever available. Unlike the stable ODE levels observed in Figure 3A–D, four patients showed increases in plasma ODEs ([CD9 on MOG], Figure 3F), another four showed substantial decreases (Figure 3G), and only two remained stable (Figure 3F). Among the 11 patients, one ovarian cancer patient receiving paclitaxel and carboplatin developed clear cognitive decline (chemo brain) (Figure 3G, arrow). Since these were gynecological cancer patients, no routine brain MRI was performed. The chemo brain patient (Figure 3G, arrow) declined MRI evaluation at the time of chemo brain diagnosis and subsequently passed away. Interestingly, three other patients whose ODE levels showed substantial decreases (Figure 3G) did not exhibit cognitive decline, underscoring the complexity of cognitive function. More detailed clinical information is available upon request.

## 4. Discussion

This study establishes a new framework for assessing white matter by measuring ODEs in blood. ODEs were clearly visualized under SEM (Figure 1), and the assay was validated (Figure 1 and Figure 2). The approach represents a shift from our earlier work on neuron-derived EVs (NDEs) [20]. Although neurons have long been the primary focus in neurology, their heterogeneity across brain regions and neurotransmitter systems has complicated translation into clinical assays. By contrast, OLs constitute a relatively homogeneous, well-defined population distributed throughout white matter. Targeting ODEs therefore provides a more stable and interpretable signal, offering a complementary perspective to the neuron- and astrocyte-focused biomarkers that dominate current research.

Numerous review articles have highlighted the value of EVs for evaluating white matter [21,22,23]; however, the use of circulating ODEs in human blood remains limited [24,25]. Agliardi et al. [24] employed anti-MOG antibodies similar to ours, while Yu et al. [25] used anti-CNPase (2’,3’-cyclic nucleotide 3’-phosphodiesterase) to isolate ODEs. In both studies, the captured ODEs were lysed for downstream analyses. We had conducted similar experiments for over a decade prior to the first publication on circulating NDEs [20] and observed that the three-step process (EV capture, lysis, and protein assay) introduced variability at each stage. To address this, we developed a simple, one-step sandwich assay that maintains EVs intact, using an EV-specific anti-CD9 antibody and an oligodendrocyte-specific anti-MOG antibody. According to our knowledge, this assay is novel.

White matter has often been treated as the brain’s “black box,” with damage detected only after symptoms arise. Conventional MRI identifies only gross abnormalities, while advanced fMRI and dMRI provide valuable research tools but are not widely available in routine practice. No current imaging or molecular biomarker directly visualizes white matter injury at a microstructural level. The ODE assay thus provides a biological window into this otherwise inaccessible compartment. While clinical gold standards are lacking, proof-of-concept validation was achieved through archived samples from clearly defined neurological events—including head trauma, stroke, and chemotherapy exposure (Figure 3). These diverse cohorts, though heterogeneous in size, gender, age, and timing of blood collection, underscore the assay’s robustness and highlight its suitability for future controlled clinical trials.

Increases in ODE levels may reflect OL activation and the initiation of repair, whereas sustained decreases may indicate significant OL damage or loss. However, interpretation is far more complex. EVs are secreted into the interstitial fluid, diffuse through extracellular matrix, and enter the bloodstream primarily via transendothelial transport [26] or lymphatic drainage [27]. Once in circulation, plasma EVs are cleared by the reticuloendothelial system in the liver, spleen, kidney, and lungs [28,29]. Thus, plasma ODE levels reflect a balance of multiple processes. In fact, under hyperlipidemic conditions, there is an increase in circulating EVs due to both increased release and decreased uptake by liver cells [30]. Changes in plasma ODE levels shown in Figure 3 do not provide conclusive information, but provide valuable insights into patient-specific conditions. In the future, if ODE data is available in real time, it could potentially be used to screen at-risk athletes and to monitor disease progression and therapeutic responses. Monitoring ODE dynamics in patients may facilitate precise, personalized management of neurological health.

Clinical endpoints such as improvements in motor, sensory, and cognitive functions primarily depend on the condition of the corresponding gray matter regions in the brain and do not directly reflect white matter integrity. When white matter integrity is compromised, signal conduction becomes slowed or uncoordinated. Therefore, once at-risk individuals are identified, white matter function can be evaluated using neuropsychological and physiological assessments such as the Trail-Making Test [31], vestibular reaction time [32], and gait analysis [33]. Although complete white matter repair remains elusive, disease progression can often be slowed through risk management, lifestyle modification, experimental interventions [1,2,3,4,5,6,7], or emerging approaches such as brain–computer interfaces [34]. Consequently, a sensitive and minimally invasive screening tool would fill a critical gap by enabling earlier detection, patient stratification for clinical trials, and monitoring of therapeutic responses.

Figure 3 showed preliminary data without any clinical conclusion. However, because the assay presented in this study is a simple sandwich immunoassay, it could be adapted as a laboratory-developed test (LDT) for research purposes or for certain clinical applications, pending resolution of legal and regulatory requirements. For diagnostic use in specific clinical conditions, validation through large-scale clinical trials would be necessary.

## 5. Conclusions and Future Directions

We developed and validated a blood test that quantitatively measures ODEs (Figure 1 and Figure 2), requiring only a small volume of venous or capillary blood and standard immunoassay equipment. The assay reliably detects CD9^+^MOG^+^ EVs and potentially captures dynamic changes in white matter injury over time (Figure 3). Validation of ODEs will encourage researchers to advance proteomic profiling, single-vesicle analytics, microfluidic chip development, and machine learning-based trajectory analysis. Beyond establishing technical feasibility, this work demonstrates four advances: (i) a shift in focus from neurons to OLs and white matter, (ii) a non-invasive method potentially capable of detecting changes before symptoms arise, (iii) validation of ODEs in human plasma, and (iv) feasibility data with representative clinical examples showing potential for longitudinal trajectories of disease impact.

The potential reach of this tool extends well beyond conventional neurology. Early applications may include psychiatry (e.g., substance withdrawal, mental disorders), oncology (chemotherapy-related brain injury, brain tumors), pediatrics (normal and abnormal brain development), geriatrics (brain aging), sports and military medicine (concussion, repetitive head trauma), and occupational health (toxic or environmental exposures). In parallel, the assay may aid industries developing drugs, devices, or digital platforms for brain monitoring and rehabilitation, with implications for insurance, workers’ compensation, and even legal contexts.

By transforming OL biology into a practical, minimally invasive test, this work opens a new chapter in the assessment of white matter health. Future studies will focus on prospective validation in larger cohorts, integration with imaging and neuropsychological testing, and longitudinal monitoring to clarify how ODE dynamics reflect disease mechanisms, therapeutic response, and resilience across the human lifespan.

## Figures and Tables

**Figure 1 cells-14-01771-f001:**
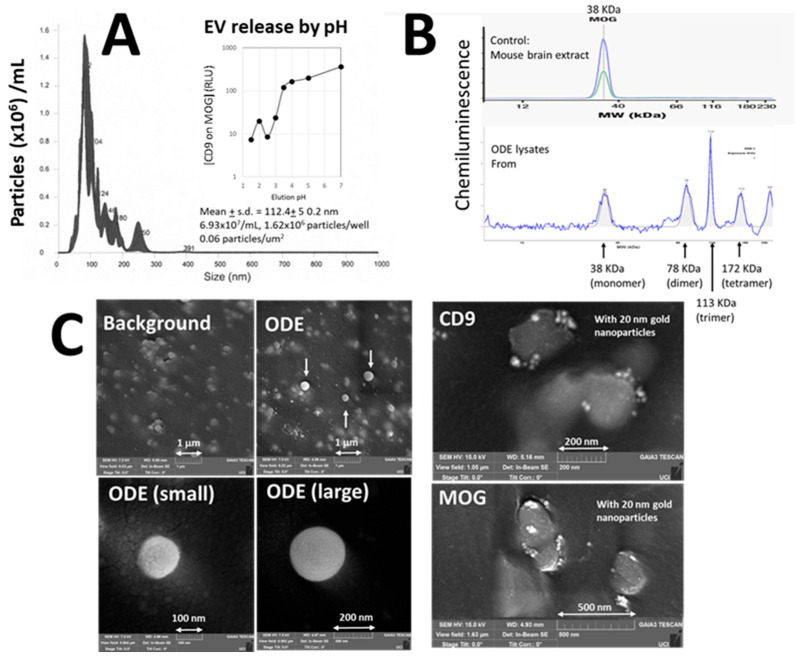
**Validation of ODEs**. (**A**) **Nanoparticle tracking analysis (NTA).** Twenty μL of pH 2.5 solution was added to the first well for 30 sec., then immediately transferred to a new well containing 3.8 µL of neutralization buffer. This process was repeated 64 times, and the pooled sample was subsequently analyzed by NTA. X-axis: particle size in nm. Y-axis: particle concentration (particles (10^6^) /mL). **[Result]** 100–200 nm particles were identified. **Inset**: EV release from ELISA wells by various pH conditions. X-axis: pH, Y-axis: ELISA readings of relative light units (RLU) of [CD9 on MOG]. **[Results]** pH < 3 was needed to elute EVs from ELISA wells. (**B**) **Western blot.** To prepare highly concentrated ODE lysates, 40 µL of sample loading buffer was added to the first well, incubated for 2 min with shaking at 700 rpm, and the supernatant was transferred to the next well. This procedure was repeated 28 times to collect a concentrated sample. Western blotting was performed using a biotinylated anti-MOG antibody specifically suitable for Western blot, distinct from the capture antibody used in the assay. Because the assay employed large amounts of mouse and rabbit IgG to block non-specific EV binding, conventional anti-MOG followed by secondary antibodies was ineffective. **Top**: 38 KDa MOG derived from mouse brain extract as a control. **Bottom**: Our ODE sample. X-axis: molecular weight. Y-axis: Chemiluminescent signals. Arrows indicate the molecular weight of monomer, dimer, trimer, and tetramer of MOG. **[Result]** MOG was identified in our ODE lysates. (**C**) **Scanning electron microscopy (SEM).** Because SEM requires a flat surface, the bottoms of ELISA wells were excised using a punch, and assays were performed in a humidity chamber to prevent evaporation. Samples were then sputter-coated with ~4 nm of iridium and imaged using a Tescan GAIA3 SEM (Czech Republic) at the UC Irvine Materials Research Institute (IMRI). **[Result] Left-middle panels**: 100–200 nm spherical particles were identified. **Right panels**: In subsequent experiments captured ODEs were incubated for 1 h with biotinylated anti-CD9 (top) and anti-MOG antibodies (bottom), followed by labeling with streptavidin-conjugated 20 nm gold nanoparticles. Gold nanoparticles were clearly seen on EVs as bright white spots.

**Figure 2 cells-14-01771-f002:**
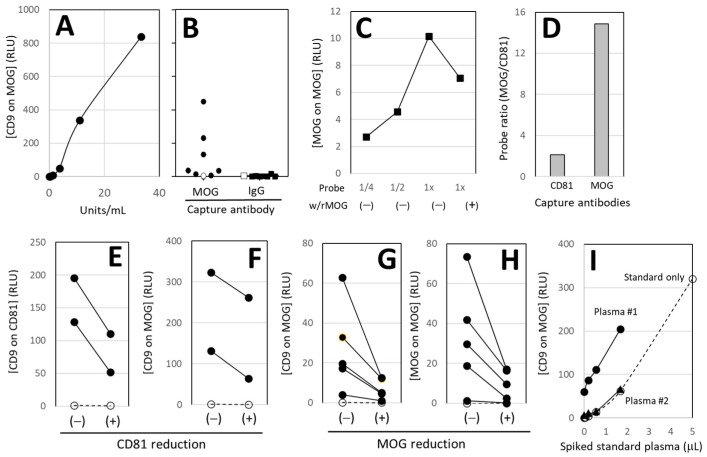
**Validation of sandwich immunoassay.** (**A**) **Standard plasma dilution.** X-axis: plasma concentration in units/mL. Y-axis: RLU of [CD9 on MOG]. **[Result]** Standard plasma showed linear dilution curve, indicating that such dilution curve could be used to convert ELISA readings of RLU to universal, units/mL. (**B**) **MOG-specificity and inter-individual variability.** Seven plasma samples (●■) and a buffer control (PBS, ○□) were applied to ELISA wells coated with either anti-MOG or control IgG. Y-axis: RLU of [CD9 on MOG]. **[Result]** [CD9 on MOG] was higher in anti-MOG well than control IgG wells, indicating MOG specificity. (**C**) **MOG-specificity.** After capturing MOG^+^ EVs on anti-MOG wells, various concentrations of anti-MOG probes were applied in the presence or absence of recombinant MOG (rMOG). **[Result]** rMOG blocked MOG reaction. (**D**) **Enrichment of MOG^+^ EVs.** Plasma samples were applied to wells coated with anti-CD81 (to capture total EVs, X-axis) or anti-MOG (to capture MOG^+^ ODE, X-axis) and probed with anti-CD81 for total EVs and anti-MOG for MOG^+^ ODE. The ratio of MOG probe/CD81 probe was shown in Y-axis. **[Result]** MOG^+^ EVs were enriched in anti-MOG wells. (**E**,**F**) **EV removal.** Two plasma samples were first applied to anti-CD81 wells to capture total EVs ((−) in X-axis), and the supernatants were subsequently transferred to second wells ((+) in X-axis), coated with either anti-CD81 (**E**) or anti-MOG (**F**). All wells were probed with anti-CD9 (Y-axis). **[Result]** Captured EVs on both anti-CD81 (**E**) and anti-MOG wells (**F**) were reduced by removing CD81^+^ EVs. (**G**,**H**) **MOG removal.** Five plasma samples were first applied to anti-MOG wells to capture MOG^+^ ODE, and the supernatants were then transferred to a second set of anti-MOG wells and probed with either anti-CD9 (**G**) or anti-MOG (**H**). **[Result]** MOG^+^ EVs ([CD9 on MOG]) were reduced by depleting MOG. (**I**) **Assay validation.** In the absence of an ideal gold standard, we used standard plasma as a tentative reference. Various volumes of standard plasma were spiked into plasma samples with moderate (#1) and low ODE levels (#2), and [CD9 on MOG] was then measured. **[Result]** Although these 2 samples had different baseline levels of [CD9 on MOG], the measured values increased proportionally with the standard plasma dilution curve, indicating that the standard plasma can serve as a reliable gold standard for the assay.

**Figure 3 cells-14-01771-f003:**
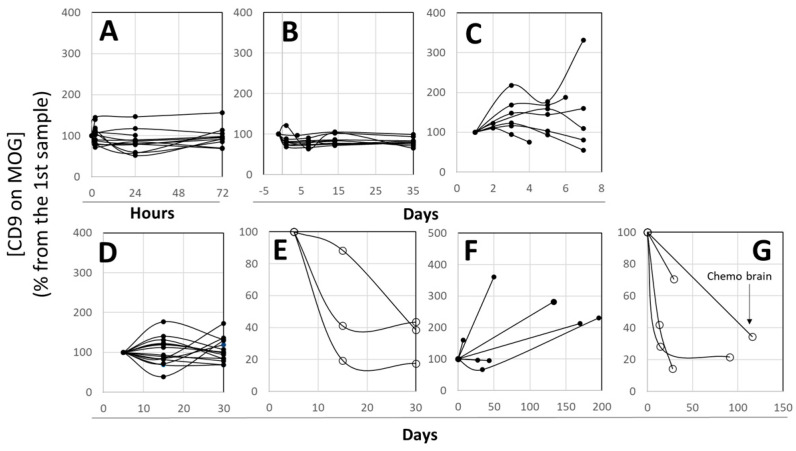
**Results of various clinical samples.** % change in [CD9 on MOG] from the 1st blood samples were shown in Y-axis. (**A**)**: Soccer heading practice**. Blood samples were collected before and at 2, 24, and 72 h after extensive heading practice. (**B**)**: Boxing and mixed martial arts (MMA)**. Blood samples were collected before and at 1, 7, 14, and 35 days after the bout. (**C**)**: Deep intracerebral hemorrhage (hemorrhagic stroke in white matter).** Blood samples were collected at 1, 3, 5, and 7 days post-stroke. (**D**,**E**)**: Subcortical (white matter) ischemic stroke.** Blood samples were collected at 5, 15, and 30 days post-stroke. D: Stable patients (*n* = 15; 8 male, 6 female; mean age 67 ± 10.6 years). E: Three patients showing a substantial decrease in [CD9 on MOG] (2 male, 1 female; mean age 65 ± 4.4 years). (**F**,**G**)**: Post-chemotherapy follow-up.** Plasma samples were collected from cancer patients (*n* = 11, all female; 6 ovarian, 4 breast, 1 vaginal; age 39–83 years, mean 64 ± 14) before and after chemotherapy. Arrow indicates a patient diagnosed with dementia.

**Table 1 cells-14-01771-t001:** Materials and methods.

Conditions	Sample Sources	Blood Collection	Gender	Age (y.o.)
Soccer heading practice	Indiana Univ.	0, 2, 24, 72 h	15 M	27 ± 7.5
Boxing and MMA	NanoSomiX	0, 1, 7, 14, 35 days	10 M	18–26 [16]
Hemorrhagic stroke	Duke Univ.	1, 3, 5, 7 days	2 M, 5 F	57 ± 7.7
Ischemic stroke	Georgetown Univ.	1, 15, 30 days	8 M, 6 F	67 ± 10.6
Cancer chemotherapy	Kawasaki Med. School	0, upto 196 days	11 F	64 ± 14
		0: pre-event		

## Data Availability

Research data are available upon reasonable request.

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
