# Peer review of "WHITE MATTER MATTERS: New Approach to the Brain’s Hidden Half Using Circulating Oligodendrocyte-Derived Extracellular Vesicles"

_cells, 2025, doi:10.3390/cells14221771_

Round 1

Reviewer 1 Report

Comments and Suggestions for Authors

Mitsuhashi et al. proposed a novel blood test that evaluates brain white matter by measuring oligodendrocyte-derived extracellular vesicles (ODEs), suggesting it may serve as a non-invasive biomarker for early detection of white matter injury. However, several issues still need to be addressed:

  1. Has the use of extracellular vesicles (EVs) as a method for evaluating white matter been previously reported in the literature?
  2. What are the specific mechanisms and underlying reasons for the increase or decrease in ODE levels?
  3. The sample size in the clinical study is relatively small. Does the author have plans to increase the sample size and establish a healthy control group?
  4. Are there any clinical imaging data available that could be correlated with and support the authors’ conclusions?
  5. The authors propose ODE levels as a marker for assessing white matter health. What are the current limitations of this approach? The authors should elaborate on this point in the Discussion section.

Author Response

Comment 1: Has the use of extracellular vesicles (EVs) as a method for evaluating white matter been previously reported in the literature?

Response 1: Please see the 2nd paragraph in the revised Discussion. Numerous review articles have highlighted the value of extracellular vesicles (EVs) for evaluating white matter (21–23); however, the use of circulating oligodendrocyte-derived EVs (ODEs) in human blood remains limited (24–25). Agliardi et al. (24) employed anti-MOG antibodies similar to ours, while Yu et al. (25) used anti-CNPase (2',3'-cyclic nucleotide 3'-phosphodiesterase) to isolate ODEs. In both studies, the captured ODEs were lysed for downstream analyses. We had conducted similar experiments for over a decade prior to the first publication on circulating neuron-derived EVs (20) and observed that the three-step process (EV capture, lysis, and protein assay) introduced variability at each stage. To address this, we developed a simple, one-step sandwich assay that maintains EVs intact, using an EV-specific anti-CD9 antibody and an oligodendrocyte-specific anti-MOG antibody. According to our knowledge, this assay is novel.

  1. Kumar A, Nader MA, Deep G. Emergence of Extracellular Vesicles as "Liquid Biopsy" for Neurological Disorders: Boom or Bust. Pharmacological Reviews. 76:199-227 (2024).
  2. Wang X, Yang H, Liu C, Liu K. A New Diagnostic Tool for Brain Disorders: Extracellular Vesicles Derived From Neuron, Astrocyte, and Oligodendrocyte. Frontiers in Molecular Neuroscience. 16:1194210 (2023).
  3. Manolopoulos A, Yao PJ, Kapogiannis D. Extracellular Vesicles: Translational Research and Applications in Neurology. Nature Reviews. Neurology. 21:265-282 (2025).
  4. Agliardi C, Guerini FR, Zanzottera M, et al. Myelin Basic Protein in Oligodendrocyte-Derived Extracellular Vesicles as a Diagnostic and Prognostic Biomarker in Multiple Sclerosis: A Pilot Study. International Journal of Molecular Sciences. 2023;24(1):894. doi:10.3390/ijms24010894.
  5. Yu Z, Shi M, Stewart T, et al. Reduced Oligodendrocyte Exosome Secretion in Multiple System Atrophy Involves SNARE Dysfunction. Brain : A Journal of Neurology. 2020;143(6):1780-1797. doi:10.1093/brain/awaa110.

Comment 2: What are the specific mechanisms and underlying reasons for the increase or decrease in ODE levels?

Response 2: Please see the 4th paragraph in Discussion:

Increases in ODE levels may reflect OL activation and the initiation of repair, whereas sustained decreases may indicate significant OL damage or loss. However, interpretation is far more complex. EVs are secreted into the interstitial fluid, diffuse through extracellu-lar matrix, and enter the bloodstream primarily via transendothelial transport (26) or lymphatic drainage (27). Once in circulation, plasma EVs are cleared by the reticuloendo-thelial system in the liver, spleen, kidney, and lungs (28–29). Thus, plasma ODE levels re-flect a balance of multiple processes. In fact, under hyperlipidemic conditions, there is an increase in circulating EVs due to both increased release and decreased uptake by liver cells (30). Changes in plasma ODE levels shown in Fig. 3 do not provide conclusive in-formation, but provide valuable insights into patient-specific conditions. In the future, if ODE data is available in real time, it could potentially be used to screen at-risk athletes and to monitor disease progression and therapeutic responses. Monitoring ODE dynam-ics in patients may facilitate precise, personalized management of neurological health.

Comment 3: The sample size in the clinical study is relatively small. Does the author have plans to increase the sample size and establish a healthy control group?

Response 3: Yes. We have already submitted some grant applications for larger studies. We are also looking for partners for subsequent clinical studies. Appropriate control group will be included.

Comment 4: Are there any clinical imaging data available that could be correlated with and support the authors’ conclusions?

Response 4: As stated in Results, MRI was only taken at the time of admission in stroke cases, and no follow-up MRI was taken. Since cancer patients used in this study was gynecological cancer, no MRI was taken. This is reasonable because these studies were conducted under the standard medical practice. Follow up MRI will be used in our subsequent studies.

Comment 5: The authors propose ODE levels as a marker for assessing white matter health. What are the current limitations of this approach? The authors should elaborate on this point in the Discussion section.

Response 5: Please see the last paragraph in the revised Discussion. Because the assay presented in this study is a simple sandwich immunoassay, it could be adapted as a laboratory-developed test (LDT) for research purposes or for certain clinical applications, pending resolution of legal and regulatory requirements. For diagnostic use in specific clinical conditions, validation through large-scale clinical trials would be necessary.

Reviewer 2 Report

Comments and Suggestions for Authors

Overall Recommendation

This is a well-conceived, innovative, and clearly written manuscript. The pilot findings are intriguing, and the approach has broad potential. I recommend publication pending minor, detail-oriented revisions focused on methodological clarification and contextual expansion.

Author Response

Comment 1: Use of acidic conditions for EV elution

Response 1: This was addressed in Fig. 1A inset.

Comment 2: As acknowledged, low pH may compromise vesicle structure. A brief discussion of potential molecular alterations and possible alternative approaches would be informative.

Response 2: Low pH is known to induce protein denaturation and conformational changes. Since monoclonal anti-MOG recognize specific structure of MOG moiety, tiny change at antibody binding site induces critical problem of antigen-antibody reaction. Exact structural change is not known. Alternative approach is to add recombinant MOG (rMOG) to replace anti-MOG binding from EV-MOG to rMOG. However, because it requires a large quantity of rMOG, it is not practical. 

Comment 3: Anti-MOG capture

As stated in 2.1. Reagents, our anti-MOG binds to extracellular domain of MOG to capture native ODEs without lysis or permeabilization

Comment 4: The rationale for targeting MOG is convincing; however, the possibility of capturing soluble MOG fragments or myelin debris could be addressed.

Response 4: Soluble MOG (sMOG) can exist in blood, but it cannot bind to anti-CD9. Since our assay only recognizes CD9+MOG+ double positive material, sMOG does not influence the assay. “Soluble MOG present in plasma does not show signals due to the lack of CD9” was inserted in the section of 3.2.1.

Comment 5: The authors might comment on future directions, such as microfluidic separation or synthetic vesicle standards.

Response 5: Regarding microfluidic separation, this sentence was inserted in the section 5.

“Validation of ODEs will encourage researchers to advance proteomic profiling, sin-gle-vesicle analytics, microfluidic chip development, and machine learning–based trajectory analysis.” see the section 5

Regarding synthetic vesicle standards, we stated in the section 3.2.6.

“It is not feasible to synthesize recombinant proteins carrying both MOG and CD9 moieties with binding characteristics identical to native MOG⁺ EVs.”

Comment 6: Interpretation (Discussions) is balanced and avoids overstated conclusions. To enhance clarity, the authors could elaborate on:

  • Mechanisms potentially underlying sustained decreases in ODE levels (e.g., oligodendrocyte loss, impaired secretion, increased clearance).
  • The dynamic balance between EV secretion and systemic clearance.
  • The interplay between injury, repair responses, and systemic physiology.

Response 6: This was stated in the 4th paragraph in the revised Discussion.

Comment 7: The inclusion of small, heterogeneous cohorts is appropriate for a feasibility study. To improve transparency please consider:

  • A summary table outlining cohort characteristics (sample size, age range, sampling times) would assist readers.

Response 7: Table I was inserted.

Comment 8: The single patient with post-chemotherapy cognitive decline should be contextualized cautiously, avoiding any impression of causality.

Response 8: In the last paragraph in section 3.3.5. stated that “Interestingly, three other patients whose ODE levels showed substantial decreases (Fig. 3G) did not exhibit cognitive decline, underscoring the complexity of cognitive function. More detailed clinical information is available upon request.”

Comment 9; Figures are generally clear and informative. Minor suggestions:

  • Subpanels in Figure 3 could be re-formatted or combined for improved readability.
  • Where possible, adding confidence intervals would assist interpretation.

Response 9: Fig.3 was replaced.

Comment 10: The reference list is current and well aligned with the scope of the manuscript, including relevant citations from the last five years.

A few additional citations on EV clearance heterogeneity and emerging EV proteomics could further strengthen the discussion.

Response 10: Regarding EV clearance heterogeneity, this paragraph was inserted in Discussion.

“In fact, under hyperlipidemic conditions, there is an increase in circulating EVs due to both increased release and decreased uptake by liver cells (30)

  1. K. Németh, Z. Varga, D. Lenzinger, T. Visnovitz, A. Koncz, N. Hegedűs, Á. Kittel, D. Máthé, K. Szigeti, P. Lőrincz, C. O'Neill, R. Dwyer, Z. Liu, E. I. Buzás, V. Tamási. Extracellular Vesicle Release and Uptake by the Liver Under Normo- And Hyper-lipidemia. Cell. Mol. Life Sci. 78:7589-7604 (2021).

Regarding proteomics, this study was focused on CD9 and MOG, not core materials. Thus, we decided not mention EV proteomics. 

Comment 11: A dedicated limitations paragraph near the end of the Discussions section would benefit future readers and reviewers.

Response 11: We added this paragraph at the end of Discussion;

“Fig. 3 showed preliminary data without any clinical conclusion. However, because the assay presented in this study is a simple sandwich immunoassay, it could be adapted as a laboratory-developed test (LDT) for research purposes or for certain clinical applica-tions, pending resolution of legal and regulatory requirements. For diagnostic use in spe-cific clinical conditions, validation through large-scale clinical trials would be necessary.”

Comment 12: IRB approval and informed consent are appropriately reported. Methodological descriptions appear sufficient to support replication.

Listing antibody lots or validation criteria would provide additional transparency.

Response 12: Validation of each antibody was fully described in Fig. 1 and 2 under our experimental condition.

Comment 13: The abstract could include the approximate number of subjects studied in each cohort for greater clarity.

Response 13: In Abstract, this paragraph was inserted.

“However, ODE levels remain stable under mild head impacts in soccer heading practice (n=15) and boxing/mixed martial arts (n=10), whereas change markedly following neurological insults such as hemorrhagic (n=7) and ischemic stroke (n=14), or gynecological cancer after chemotherapy (n=11).”

Comment 14: The Discussions section may benefit from a brief reflection on how ODE trajectories could inform clinical trial endpoints.

Response 14: See paragraph 5 in Discussion.

“Clinical endpoints such as improvement of motor, sensory, and cognitive function are dependent on the condition of corresponding gray matter lesions in the brain, not directly reflect white matter conditions. When white matter integrity is compromised, signal conduction slows or becomes uncoordinated. Therefore, white matter conditions can be assessed by neuropsychological testing, such as trail-making test (31), vestibular reaction time (32), and gait analysis (33). Although complete white matter repair re-mains out of reach, disease progression can often be slowed through risk management, lifestyle modification, experimental interventions (1–7), or novel approaches such as brain–computer interfaces (33). A sensitive, minimally invasive screening tool therefore fills a critical gap: enabling earlier detection, stratification for clinical trials, and monitoring of therapeutic response.

Comment 15: The authors may consider discussing potential integration with ODE proteomic profiling, single-vesicle analytics, machine learning-based trajectory analysis.

Response 15: In the section 5, this was inserted.

“Validation of ODEs will encourage researchers to advance proteomic profiling, single-vesicle analytics, microfluidic chip development, and machine learning–based trajectory analysis.”

Round 2

Reviewer 1 Report

Comments and Suggestions for Authors

Thank you for the authors’ response, I have no further questions.